# The Effects of Psychological Factors on Perceptions of Productivity in Construction Sites in Japan by Worker Age

**DOI:** 10.3390/ijerph17103517

**Published:** 2020-05-18

**Authors:** Nobuki Hashiguchi, Jianfei Cao, Yeongjoo Lim, Yasushi Kubota, Shigeo Kitahara, Shuichi Ishida, Kota Kodama

**Affiliations:** 1Graduate School of Technology Management, Ritsumeikan University, Osaka 567-8570, Japan; gr0401fv@ed.ritsumei.ac.jp; 2Faculty of Business Administration, Ritsumeikan University, Osaka 567-8570, Japan; lim40@fc.ritsumei.ac.jp; 3Kumagai Gumi Co., Ltd., Tokyo 162-8557, Japan; yakubota@ku.kumagaigumi.co.jp (Y.K.); skitahar@ku.kumagaigumi.co.jp (S.K.); 4School of Engineering, Tohoku University, Miyagi 980-8579, Japan; shuichi.ishida.e6@tohoku.ac.jp

**Keywords:** construction worker, age, heart rate, body mass index (BMI), structural equation modeling, Japan

## Abstract

The construction industry is a work environment that poses many dangers to workers, with many hidden factors that affect work awareness. It is important for construction companies to ensure a balance between productivity and safety in the work environment. The purpose of this study was to identify relationships between the feeling of safety in the work environment, proactive work behavior, job satisfaction, work skills, team performance, and health risk indicators, such as heart rate, among construction workers of different ages. Based on previous research, we examined the hypothetical perception model. We then administered a questionnaire survey to construction workers (*N* = 357) employed at a Japanese construction company. Using structural equation modeling (SEM), we investigated the impact of health risk indicators on worker perceptions among young and older workers. The results showed that workers’ heart rate and body mass index (BMI) had a negative effect on the feeling of safety and proactive work behavior among older workers, but showed no significant relationship among young workers. However, regardless of workers’ age, it was clear that the feeling of safety affects job satisfaction, and that work skills and proactive work behaviors affect perceptions regarding team performance.

## 1. Introduction

The construction industry is labor intensive compared to other industries, and many construction workers face workloads that exceed their individual physical capabilities [1,2]. Many construction sites are characterized by poor work environments, such as poor scaffolding, working from heights, high temperatures, and humidity. Long-term physical workloads can lead to chronic fatigue, injuries, illness, and health risks, which may in turn reduce on-site productivity [3,4,5]. The US Bureau of Labor Statistics reported that over 43,000 workers experienced fatal occupational injuries on construction sites between 2003 and 2010 [6]. In addition, studies have shown that 37.9% of US workers experience severe fatigue, leading to fatal consequences related to worker safety, health, and productivity [7]. These strict work conditions are in a similar environment at construction sites around the world [2]. According to a survey by the Ministry of Health, Labour and Welfare, 323 workers died from work-related accidents in the Japanese construction industry in 2017; these accounted for more than 30% of all industry deaths in Japan [3,8,9]. Moreover, the most frequent cause of death was falls/descent (*N* = 135) that occurred during construction work on buildings and houses. Construction is one of the most dangerous industries, given the high frequency of occupational deaths and accidents [8,10]. While occupational health and safety is a priority in the industry, it has been pointed out that Japanese construction practices do not comply with sustainability policies [11].

The construction industry serves as the industrial base in every country and is considered the biggest contributor to national economies [12]. Therefore, it is imperative for the entire construction industry to ensure the expected productivity, in consideration of working conditions and the working environment for workers, and to ensure workers’ good health and safety [13]. The primary causes of occupational accidents depend on the nature of the work, workers’ behavior, and on-site dangers in the working environment. Moreover, safety management at construction sites may be difficult to implement on a broad scale [14]. Safety measures in the working environment and workers’ awareness of occupational safety are important countermeasures against occupational accidents [15,16,17]. Given the relationship between construction workers’ age and safety, with older workers shown to be more conscious of safety and to have more professional knowledge and experience compared to young workers, studies have shown that young workers are at a higher risk of accidents than are old workers [16]. Careful concentration on work can reduce unforeseen events [18,19].

The purpose of this study was to investigate workers’ health risk indicators, such as resting heart rate and body mass index (BMI), and productivity awareness different age groups. Previous studies have examined the relationship between workers’ psychological awareness and the productivity of construction work [2,20]. However, by combining psychological factors with health risk indicators, few reports have evaluated their effects on construction workers' perceptions of productivity. 

In this study, the workers at 38 workplaces of a Japanese construction company (Kumagai Gumi) were divided into two age groups. We analyzed the relationship among workers’ health risk indicators, the awareness of feeling safe on a worksite, job satisfaction, proactive work behavior, work skills, and team performance. From a practical perspective, in Japanese construction companies where the number of older workers is growing rapidly, our research may be useful to help human resource management (HRM) understand workers’ perceptions for each age group.

## 2. Material and Methods

### 2.1. Research Background

In the consideration of a worker’s health, his or her individual characteristics, specifically BMI by heart rate, height, and weight, constitute important biometric information. Previous studies have linked heart rate variability (HRV) to physiological responses and physical activity [21]. Other studies have used the heart rate as an index to confirm physical health and activity status [22,23]. Another study used biological information, such as heart rate, to examine the relationship between office workers’ job satisfaction and work productivity [24]. Few studies have examined these relationships.

Workload is influenced by a variety of factors, including individual physical (e.g., age, exercise, and nutrition) and environmental factors (e.g., temperature, humidity, and noise) [25,26]. Various factors are known to contribute to workload [27]. However, it is difficult to determine working conditions by taking all factors into account. Therefore, measuring physical loads, based on workers’ heart rate-based measurements, can be an alternative means of understanding these factors [28,29,30,31]. Continued engagement in activities with a high physical workload results in physical fatigue among workers, causing reduced productivity and motivation, carelessness, and poor judgment, which affects work quality and job satisfaction. In addition, it is widely recognized in the construction industry that high physical workloads can cause accidents and injuries [32]. In regions such as Japan, where the working population is aging, an increase in the proportion of older workers may affect a decline in the workforce in the construction industry [33,34].

### 2.2. The Current State of the Japanese Construction Industry

The number of construction workers in Japan was at its peak in 1997, reaching 6.85 million, and has been declining since, to 5.30 million in 2018. The number of workers on construction sites decreased to 4.55 million in 1997 and 2.97 million in 2018 [35]. In addition, construction workers are mostly at an advanced age, with 34.1% found to be >55 years old in 2018, and 11.0%, <29 years old. There is an increasing need to pass on technology to the next generation and secure long-term leaders [36,37].

Figure 1 and Table 1 show the composition of construction workers in Japan and in our study by age. The two-sample Kolmogorov–Smirnov test (two-sided *p* = 0.8928) was performed to determine whether the two groups were comparable according to age. The results showed no difference between the two samples at the significance level of 5%. In addition, it could be surmised that the study participants were in the same age group as that of construction workers in Japan. The age composition and average age of the current study participants were very similar to those of workers in the Japanese construction industry.

Older workers had the best perceptions on safety, indicated the highest level of job satisfaction, and recorded the lowest accident involvement rate [16]. From a practical perspective, understanding age-related perceptions of the workplace would benefit management’s decisions regarding workers’ adaptability, work effectiveness, implementation of safety management policies, and handling of age-related accident characteristics [16,18,19]. The main objective of this study was to compare the effects of psychological factors on construction workers’ perceptions of their work across age groups. In the study, the workers were divided into two groups. Workers under the average age (≤45) were categorized as young workers and workers over the average age (≥46) were categorized as older workers.

### 2.3. The Hypothetical Model of Construction Workers

A literature review was conducted to investigate issues related to productivity through team performance within construction companies. In this study, employees’ perceptions of occupational safety were evaluated using the Work Safety Scale (WSS) [38]. Questions from the 2011 Workplace Employment Relations Study (WERS) were used to measure workers’ workplace awareness, the relationship between supervisors and workers, job satisfaction, attitudes toward work, and skills [39,40]. Through an interview process with experienced site supervisors of the surveyed construction company, we devised a study questionnaire that consisted of items on psychological factors that may affect construction workers’ perception of work, with five of these items designed to address questions relating to the work environment. 

Our self-developed questionnaire was used to assess construction workers’ perceptions regarding the suitability of their working environment. Below, we present the hypotheses that this study intended to test; we also explain the link between these and previous research, as discussed above.

#### 2.3.1. Hypothesis 1 (H1). Health Risk Indicators Affect the Feeling of Safety and Proactive Work Behavior in the Workplace

In this study, the health risk indicators of workers were measured by the resting heart rate and BMI. The construction industry is subject to strict work schedules, irregular working hours, a need for a controlled air temperature in the work environment to ensure workplace safety, and workers being subject to poor work performance as a result of work stress [41]. To investigate the effects of a hot working environment in construction sites and the health of workers, and to reduce workers’ heat stress in the workplace, strict management of the working environment and the management of workers’ health are necessary. Previous studies have shown the importance of reducing the impact on workers’ health [42,43]. 

An elevated resting heart rate has an impact on mortality risk, heart disease, and cardiovascular disease [44,45]. In a study of a cohort of East Asians [35], people with a BMI (weight divided by height squared) in the range of 22.6 to 27.5 had the lowest risk of death. The higher the BMI, the higher the risk of death from cancer, cardiovascular disease, and other causes.

Construction workers care about their health and well-being at work [46] and recognizing their daily health risks has increased expectations for working conditions, which is consistent with increased sensitivity to safety and positive work. As part of an annual health check [47], the construction company regularly measures employees’ physical characteristics, such as heart rate, blood pressure, height, and weight, in order to increase their health and health care awareness, including providing workers with their heart rate, weight, and BMI measurements. Workers’ perceptions of health risks can lead to expectations and positive behaviors about their working conditions.

#### 2.3.2. Hypothesis 2 (H2). A Feeling of Safety in the Workplace Affects Job Satisfaction

Overload and poor working conditions reduce workers’ job satisfaction [48]. Job satisfaction depends on a variety of factors, but strategies to improve it depend on work conditions [49]. Mclain focused on workers’ stress related to their perceptions of work environments and showed that individuals are regularly exposed to health and safety threats, and that subjective interpretations of risk can affect job satisfaction [50]. Analysis of job satisfaction and perceptions of safety show that workers with high job satisfaction have a positive view of safety in the work environment and that those who are dissatisfied have a negative view of safety in that environment [51]. Other studies have shown the relevance of both work satisfaction and perceived environmental safety [52,53]. Workers can work without worry and anxiety when they have a sense of security in the work environment and high work satisfaction.

#### 2.3.3. Hypothesis 3 (H3). Job Satisfaction in the Workplace Affects Work Skills

There is no general agreement on defining job satisfaction or occupation yet [54]. Job satisfaction is a complex and multifaceted concept that has different meanings according to different authors. Job satisfaction is usually related to motivation, but it is not the same as motivation and may be related to personal feelings of achievement [55]. Workers’ job satisfaction is correlated with the breadth of their work abilities, which play an important role in the workplace [56], the congruency between workers’ required and actual skills [57], and workers’ skill development [58]. 

In most of the literature on job satisfaction, there are dealings with the effects of explanatory variables such as wages, working hours, education, health status, contract type, and work skill [57]. Matching worker expectations with the actual situation at work affects job satisfaction. This study considers matches in job satisfaction, and job skills which is one of the work aspects. Bos et al. noted that inconsistencies between actual work skills and work requirements can have a negative impact on work life and therefore affect the relationship between work skills and job satisfaction [59]. Workers with high job satisfaction are likely to have higher professional and technical skills related to construction machines and specialized equipment at the workplace, resulting in more productivity.

#### 2.3.4. Hypothesis 4 (H4). Team Performance in the Workplace Is Affected by Workers’ Work Skills, Feeling of Safety, and Proactive Behavior

##### Hypothesis 4-1 (H4-1). Work Skills Affect Team Performance in the Workplace

Improvement in work skills, such as workers’ technical abilities and equipment proficiency, is central to increasing worker productivity in the workplace, maintaining productivity growth, and translating that growth into more and better work [60]. Previous studies have quantitatively reported that inadequate job skills reduce worker productivity [61], and that a one-point reduction in work capacity can reduce employee productivity by up to 5% [62]. Using the Job Demands-Resource (JD-R) model, Baik and colleagues pointed out Hypothesis 4-1 (H4-1), in which workforce employees are likely to become enthusiastic about their jobs when applying job skills [63].

##### Hypothesis 4-2 (H4-2). The Feeling of Safety Affects Team Performance in the Workplace

Problems occurring in workplaces where there is a risk of accidents can easily lead to the actual occurrence of serious work accidents, which could result in workers taking time off, and to a reduction in workplace productivity [64]. Safety and productivity are closely linked, and employers need to pay attention to these in the work environment [49]. Perceptions of safety in the work environment have been found to correlate with worker health and accident frequency, and workplace safety management and safety practices have been found to significantly correlate with workplace accident rates [19,65]. Workers’ feeling of safety in their work environment can reduce work stress and psychological and physical dissatisfaction and may increase work satisfaction and awareness of workplace productivity.

##### Hypothesis 4-3 (H4-3). Proactive Work Behaviors Affect Team Performance in the Workplace

Proactive work behaviors relate to making things happen and being actively productive at work. Positive behavior at work is beneficial for personal and business success [66,67]. Work performance is not just the performance of work tasks, as an active approach towards work tasks is also important [68]. Positive behavior in the workplace affects job performance and worker well-being, leading to team effectiveness and organizational performance [69]. In addition, a workplace that actively fosters employees’ interpersonal relationships is beneficial to employee attitudes, organizational performance, and workers’ careers, and reduces turnover and improves performance [70]. 

##### Hypotheses 4-1–4-3

As a result, work skills (H4-1), feeling of safety (H4-2), and proactive work behaviors (H4-3) increase workers’ productivity through teamwork. The awareness of team performance means that workers’ awareness leads to team effectiveness and organizational performance [69]. In assessing team performance, although there are available many models of team performance, there is no one-size-fits-all approach, and it must be tailored to its environment [71]. It is important for a highly productive team to engage in activities among team members, rather than relying solely on task operations such as interacting with tools and systems [72]. In this study, team activities, such as cooperation and communication among team members, are important in recognition of team performance.

#### 2.3.5. Structural Model and List of Variables

For this study, we developed a hypothetical structural model for a workers’ perceptions model involving six latent variables, based on previous studies, past empirical knowledge, theoretical expectations, and evaluations by a co-researcher. To analyze the relationship between workers’ health risk indicators and awareness, the hypothetical model of this study is shown in Figure 2. The definition of the latent variables is shown in Table 2.

### 2.4. Data Collection

We analyzed the effects of, and relationships between, self-awareness, occupational safety awareness, productivity awareness, and work awareness in construction work, using data obtained from questionnaires completed by construction workers, as well as the workers’ resting heart rate and BMI. Structural equation modeling (SEM) was used to analyze the questionnaire data, while confirmatory factor analysis (CFA) was used to analyze the relationship between the structure of the hypothetical model and observed variables. R Language 3.5.1 was used as the programming language and development environment for statistical analysis; R Studio was used as the development environment.

In the construction company that participated in the study, the workers regularly undergo medical check-ups and physical assessments and they also have an opportunity to take note of their own measured values. Other than age, participants were required to provide physical information such as height and weight, and biological information regarding resting heart rate. A 5-point Likert scale was used for responses on the Work Environment Awareness Questionnaire to enable assessment of the impact of work environment awareness on latent variables in construction workspaces. Participants were required to respond to questions relating to feelings of safety, proactive work behavior, job satisfaction, work skills, and team performance (Appendix A). 

We explained the purpose of this research to all participants before the questionnaire. The participant's rights (participation in the questionnaire did not cause any disadvantage to the participants, ensuring that the collected data was anonymous, and that the participants could be interrupted during the questionnaire response) were explained to the participants. Thirty-three participants who did not respond to any of the questionnaire items were not included in the analysis. The total number of participants used in the analysis was 324, with 165 young workers aged ≤45 years, and 159 older workers aged ≥46 years. Participants’ demographics, such as gender, age, employment status, and employment level are shown in Table 3. The composition of either worker group is shown in Table 4.

### 2.5. Data Analysis

We compiled the questionnaire results related to the observed variables that affected each latent variable in the virtual model in Figure 2 and performed statistical analysis on Group A (young workers) and Group B (older workers).

Prior to analysis of the questionnaire responses through SEM, the reliability of the responses was analyzed using CFA [73]. In a hypothetical structural model, observed variables with a weak factor loading for the latent variables are excluded [74]. Groups A and B initially consisted of 17 observed variables, but after analysis, seven observed variables did not show a strong factor load on the latent variables in both groups. As a result, the number of variables was reduced to 10, as weak factor loadings were eliminated [75]. In both groups, the deleted observed variables in feeling of safety is FS3; proactive work behaviors are PWB3 and PWB4; job satisfaction is JS3; work skills are WS3; and team performance are TP3 and TP4 (Appendix A). Before further analysis, the measurement properties were examined to ensure the model’s reliability and validity [76]. First, to check the reliability and validity of the data, we obtained Cronbach’s alpha [75]. A value greater than 0.7 is generally considered ideal to ensure data reliability [77]. The reliability values for all constructs are shown in Table 5. All values were above 0.7, indicating good confidence for both datasets; these datasets were subsequently used for analysis.

The BMI and resting heart rate obtained from the worker’s answer were used as the observed variables for investigating the effect of the latent variable health risk index (HRI). In Groups A and B, HRI analyzes the relationships between the other five latent variables by SEM.

The descriptive statistics and correlation results for the young workers are shown in Table 6. The latent variables with the highest ratings were productive work behaviors (PWB; mean = 3.99, standard deviation (SD) = 0.86), followed by team performance (TP; mean = 3.94, SD = 0.90), and feelings of safety (FS; mean = 3.65, SD = 0.94). The average scores for job satisfaction (JS) and work skills (WS) were 3.31 and 3.08, respectively. HRI did not yield any statistically significant correlations with other latent variables, and WS did not show significant correlation with FS and PWB. With the exception of these, the remaining variables were positively correlated with each other. Correlations were found between JS and WS (r = 0.21, *p* < 0.01), between FS and TP (r = 0.26, *p* < 0.01), and between WS and TP (r = 0.27, *p* < 0.01). The correlation between FS and JS (r = 0.49, *p* < 0.01), and that between PWB and TP (r = 0.41, *p* < 0.01) constituted the largest correlations among the latent variables.

The descriptive statistics and correlation results for older workers are shown in Table 7. The latent variables with the highest ratings were PWB (mean = 3.84, SD = 0.82), followed by TP (mean = 3.80, SD = 0.89), and FS (mean = 3.59, SD = 0.87), similar to the results for young workers. The average scores for WS and JS were 3.44 and 3.21, respectively. HRI showed a negative correlation with FS and PWB, however the correlation between HRI further PWB was not significant. In addition, other latent variables that showed a positive correlation with HRI were not significant. The relationship between JS and PWB was not significant. With the exception of these, all variables were positively correlated and significant. The correlation between FS and JS (r = 0.39, *p* < 0.01) and that between JS and WS (r = 0.48, *p* <0.01) constituted the largest correlations among the latent variables. Correlations were found between WS and TP (r = 0.22, *p* < 0.01), between FS and TP (r = 0.29, *p* < 0.01), and between PWB and TP (r = 0.25, *p* < 0.01).

## 3. Results

SEM was used to analyze the results of the questionnaire survey on construction workers. This analysis method constructs hypotheses about causal relationships between items or variables and verifies observed variables, such as responses to questionnaires, test scores, and experimental data. SEM is a statistical approach used to understand social and natural phenomena by introducing latent variables that cannot be directly observed, and for identifying the causal relationships between the latent variables and the observed variables. For the measured model, a series of regression analyses (pass analysis) is included in the same model and analyzed simultaneously. Specifically, a hypothesis regarding the assumed causal effect is modeled, and the strength and direction of the causal relationship are estimated and tested while examining the validity of the obtained hypothesis model and modifying the hypothesis model. Regression analysis and path analysis do not take measurement errors into account (i.e., the assumption is that there is no measurement error in the model) and use regression analysis and path analysis to confirm that the assumptions are not invalid. This is followed by evaluation of the measurement model. In this paper, a path diagram is shown in Figure 3 and Figure 4 with a standardized solution to find the strength of the path coefficient [78].

In order to investigate the relationship between latent variables, analysis was performed for both Group A and Group B models using SEM. We compared the relationships between workers’ perception and latent variables across the two groups. Group A and Group B were analyzed separately to examine the relationship between the predicted direction and latent variables. The awareness model of young workers is shown in Figure 3, and that of older workers is shown in Figure 4. The direct and indirect effects of all latent variables have been measured. The path coefficient in the hypothesis model indicates the estimated value of the standardized parameter.

In the young worker model shown in Figure 3, the hypothesis from HRI (health risk indicators) to FS (feeling of safety in the work environment) and PWB (proactive work behaviors in the workspace) was not significant, while the other hypotheses were confirmed. In the model for older workers shown in Figure 4, the direction from HRI to FS and from HRI to PWB had a negative effect, whereas the direction between the other latent variables had a positive effect. The path coefficients were significant, confirming that H1 until H4-3 hypotheses were correct in the model of older workers.

In the hypothesis model, HRI had a direct effect on FS and PWB, and an indirect effect on TP (perception of team performance). Group A and Group B results show that, regardless of workers’ age, FS and PWB have a direct impact on TP, and that HRI of old workers indirectly affect TP.

We verified the fitness of each model for Group A and Group B. We also checked whether the values of the goodness of fit (GoF) index were satisfactory and evaluated the validity of the measurement model [79]. Measures of goodness of fit were the Normkai square (X^2^/df), comparative fit index (CFI), Tucker–Lewis index (TLI), goodness of fit index (GFI), and the root mean square error of approximation (RMSEA). GoF indices for both models were obtained according to the recommended model fit index [20,79,80,81]. The GoF indices for both groups (Group A and Group B) fit the recommendations, and the recommended values are shown in Table 8. The hypothetical model had a good fit for both groups.

Groups A and B were analyzed to determine the relationship between predicted direction and latent variables. As shown in Table 9, the direct and indirect effects of all latent variables were measured. In the hypothesis model in this paper, health risk indicators were shown to have a direct impact on safety awareness and active work behavior. According to the results for the two groups, not only did older workers’ health risk indicators negatively affect the feeling of safety and proactive work behaviors, but their job satisfaction and work skills were also shown to have a negative impact on team performance. The relationship that the above health risk indicators affect other latent variable was identified only in the older group of construction workers; health risk indicators had no significant effect on other latent factors in the young group. Health risk indicators were shown to affect perceptions of team performance only among older workers. The above discussion indicates differences between young and older workers.

## 4. Discussion

On a labor-intensive construction site, it is very important for construction companies to understand workers’ awareness of safety and productivity. The factors discussed in this study include psychological factors that are closely related to the working environment in the Japanese construction industry. The aim of this study was to investigate the relationship among health risk indicators, such as resting heart rate and BMI, and working awareness about safety and productivity in construction workers. The results show that health risk indicators do not affect young workers’ awareness of team performance in workplace, but that it does affect those of older workers. Among older workers, all the hypotheses used in this study are consistent with previous studies. For older workers, the health risks indicators suggest that they are concerned for safety and proactive work behaviors [46,47]. Regardless of workers’ age, a feeling of safety affects awareness of team performance through job satisfaction and work skills [48,50,52,53,56,60], and proactive work behaviors affect awareness of team performance [70,79]. Team performance is associated with productivity in the workplace [70], it becomes an important management theme at the construction site.

Moreover, the impact of a feeling of safety on team performance was slightly greater for older workers compared to young workers. Previous studies have reported that older construction workers have reduced abilities to work, as the age increases [46]. It is similar to the findings of not only blue-collar workers but also other workgroups such as white-collar workers, commercial service workers, and home care workers [84,85,86]. As construction workers grow older, they feel more anxious about their own health [46], and so their feeling of safety about working environment and awareness of proactive work behaviors may decrease, as shown in the above previous research. Conversely, there was no effect on younger workers' awareness about the feeling of safety at working environment and proactive work behavior.

In contrast, the impact of proactive work behaviors on perceived team performance in the workplace was higher for young workers than for old workers. This result shows that young workers are more aware of proactive work behavior and team performance. Crawford et al. reviewed a number of publications and found that there were some physical and psychological changes related to age [87]. In addition, young workers have higher physical and mental abilities than older workers and want to acquire new technical skills [88]. However, these studies only address some capabilities, and there is generally a negative stereotype for older workers, although a few previous studies have dealt with changes in proactive work behaviors and team performance according to age.

In addition, regardless of age, a feeling of safety was found to have a direct impact on job satisfaction and an indirect impact on work skills. This indicates that job safety affects job satisfaction [48,49,52,53,58]. In this study, job satisfaction and work skills were used as mediators of the relationship between feeling of safety and awareness of team performance.

Although the difference between young and older workers was slight, this study confirmed that health risk indicators for older workers, but not young workers, are affected by other psychological factors at the construction site**.**

## 5. Conclusions

In this study, we conducted a questionnaire survey on safety awareness in construction work among workers from two age groups in a Japanese construction company. The health risks associated with a high resting heart rate among young and older workers have been shown to have a negative effect on workplace productivity and safety awareness in construction work. In particular, regular monitoring of biological information, such as heart rate, and physical information, such as BMI, can help facilitate an enduring work relationship between construction companies and workers by fostering a sense of awareness among older workers. Our study found that health risk indicators, feeling of safety, job satisfaction, awareness of work skills, and proactive work behaviors in the workplace are suggested to affect workplace productivity. By having a quantitative understanding of workers’ health risk and awareness, construction companies can improve workers’ Qualty of Work (QoW) and well-being.

Construction projects involve several daily business problems, and therefore HRM is important in building sustainable development projects. At the construction site, the workforce experiences changes as the project progresses, and the different skills, experiences, and health and safety risks of several workers must be managed. Previous studies on these workers have conducted interviews and questionnaire surveys and created research reports based on HRM [89,90,91,92], but few have analyzed biological information and physical characteristics. Further research on this can facilitate management of construction site productivity through teamwork by gaining an awareness of workers.

The findings of this study have several limitations. First, the study used a self-reporting method that could result in differences between workers’ provided and actual resting heart rate, height, and weight. Future research should seek to include observable data to better understand the potential consequences of workers. Second, the study focused on workers at a Japanese construction company and infers causality in their awareness. Organizational activities at Japanese construction sites are characterized by high levels of worker empowerment, continuous improvement through teamwork, and a positive attitude towards quality and safety [93,94]. Including these strong points of the Japanese construction industry, future studies may need to investigate more construction workers, including factors such as organizational culture, job engagement, and job crafting behavior, to confirm the reported results. Third, worker perceptions are very complex because they are influenced by factors other than age, such as previous experience, physical characteristics, and private situations, which further research should address. Last, the model hypothesis used in this study is diverse, with many side effects and adverse effects, as well as endogenous problems. The suggested results possibly depend on a case study of workers in a Japanese construction company, and the extent of generality of the results needs further discussion.

## Figures and Tables

**Figure 1 ijerph-17-03517-f001:**
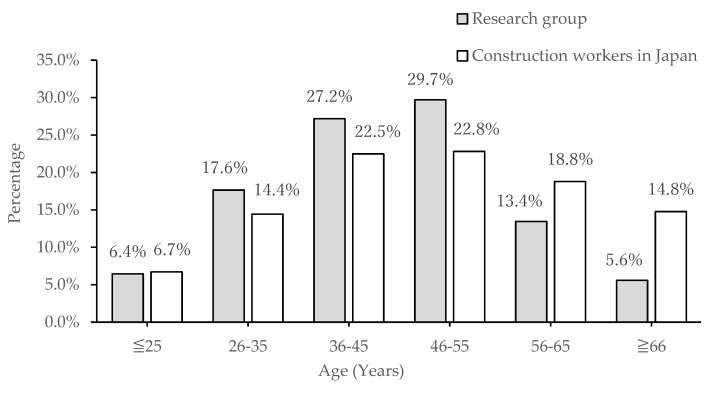
Comparison of construction workers by age group.

**Figure 2 ijerph-17-03517-f002:**
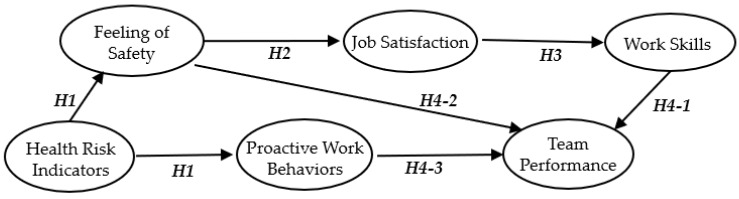
Hypothetical structural model for construction workers’ perceptions.

**Figure 3 ijerph-17-03517-f003:**
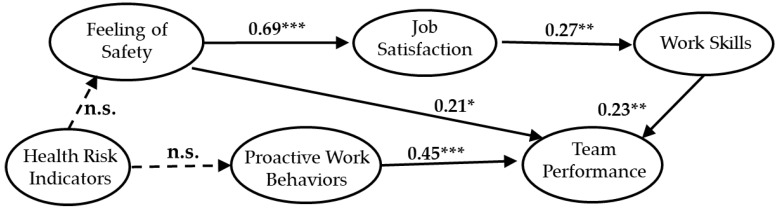
Standardized parameter estimates for the final structural model of young workers. * indicates *p* < 0.05; ** indicates *p* < 0.01; *** indicates *p* < 0.001; n.s.: not significant. The path coefficient is a standardized estimate value.

**Figure 4 ijerph-17-03517-f004:**
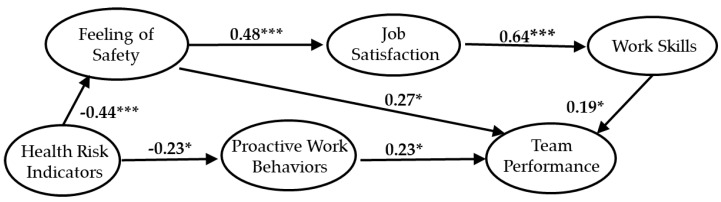
Standardized parameter estimates for the final structural model of older workers. * indicates *p* < 0.05; *** indicates *p* < 0.001. The path coefficient is a standardized estimate value.

**Table 1 ijerph-17-03517-t001:** Age group frequency across the research group and the construction worker population in Japan.

Age (Years)	Participants		Construction Workers in Japan	
	Frequency	Percentage	Frequency (×10^4^)	Percentage
≤25	23	6.4	20	6.7
26–35	63	17.6	43	14.4
36–45	97	27.2	67	22.5
46–55	106	29.7	68	22.8
56–65	48	13.4	56	18.8
≥66	20	5.6	44	14.8
Total	357	100.0	298	100.0
Average age (years)	44.8		45.6	

**Table 2 ijerph-17-03517-t002:** The definition of the variables used in this study.

Variable	Definition	Source References
Health Risk Indicator	Workers in a strict environment may be concerned about their own health. The health risk indicators of workers are measured by the resting heart rate and BMI.	[2,35,44,45]
Feeling of Safety	Workers’ feelings of safety about the workplace environment can reduce work stress, psychological and physical dissatisfaction.	[19,65]
Proactive Work Behaviors	Proactive work behaviors are about making things happen at work. This includes the potential to shape employee proactivity through designing work structures, leader behaviors, and work climates that foster employees’ confidence, activate challenging goals, and promote positive effects.	[70]
Job Satisfaction	Workers’ job satisfaction is correlated with the breadth of their work abilities, and work skills play an important role in the workplace. In addition, the congruency between required work skills and a worker’s actual work skills is related to job satisfaction.	[56,57]
Work Skills	Improvement in work skills, such as workers’ technical abilities and equipment proficiency, increase worker productivity in the workplace. Skills development is central to improving productivity and helps to maintain productivity growth and translate that growth into more and better work.	[60]
Team Performance	Workers’ awareness leads to team effectiveness and organizational performance. Team activities, such as cooperation and communication among team members, are important in recognition of team performance.	[69,72]

Note: BMI: Body Mass Index.

**Table 3 ijerph-17-03517-t003:** Participant demographics (*N* = 324).

Characteristics	Category	Frequency	Percentage
Gender	Male	322	99.4
	Female	2	0.6
Age (years)	≤25	21	6.6
	26–35	60	18.5
	36–45	84	25.9
	46–55	95	29.3
	56–65	46	14.2
	≥66	18	5.6
Experience (years)	≤10.0	112	34.6
	10.1–20.0	85	26.2
	20.1–30.0	83	25.6
	30.1–40.0	31	9.6
	≥40.1	13	4.0
Employment level	Worker/Helper	15	4.6
	Engineer/Technician	263	81.2
	Supervisor/Manager	46	14.2
Part-time/Full-time	Part-time	0	0.0
	Full-time	324	100.0

**Table 4 ijerph-17-03517-t004:** Participants’ demographic characteristics (*N* = 324).

Group	Total	Percentage	Characteristics		Frequency
Young workers (Group A)	165	50.9	Age (years)	≤25	21
				26–35	60
				36–45	84
			BMI	≤22.5	63
				22.6–27.5	79
				≥27.6	23
			Heart Rates (beats/min)	≤69	52
				70–79	83
				80–89	29
				90–99	1
				≥100	0
Older workers (Group B)	159	49.1	Age (years)	46–55	95
				56–65	46
				≥66	18
			BMI	≤22.5	58
				22.6–27.5	74
				≥27.6	27
			Heart Rates (beats/min)	≤69	0
				70–79	46
				80–89	90
				90–99	23
				≥100	0

**Table 5 ijerph-17-03517-t005:** Validity and reliability tests of both worker groups.

Variables	Group A				Group B			
	Initial		Final		Initial		Final	
	Item	α	Item	A	Item	A	Item	A
Questionnaire	17	0.86	10	0.79	17	0.85	10	0.79
FS	3	0.55	2	0.72	3	0.71	2	0.82
PWB	4	0.81	2	0.81	4	0.76	2	0.92
JS	3	0.68	2	0.70	3	0.74	2	0.70
WS	3	0.74	2	0.93	3	0.82	2	0.82
TP	4	0.82	2	0.80	4	0.71	2	0.73

Note: FS: feeling of safety; PWB: proactive work behaviors; JS: job satisfaction; WS: work skills; TP: team performance.

**Table 6 ijerph-17-03517-t006:** Descriptive statistics and correlation matrix for young workers.

Variables	Mean	SD	HRI	FS	PWB	JS	WS	TP
HRI	–	–	1.00					
FS	3.65	0.94	−0.04	1.00				
PWB	3.99	0.86	−0.03	0.25 **	1.00			
JS	3.31	0.90	−0.02	0.49 **	0.29 **	1.00		
WS	3.08	0.91	0.10	0.12	0.12	0.21 **	1.00	
TP	3.94	0.90	0.08	0.26 **	0.41 **	0.33 **	0.27 **	1.00

Note: HRI: health risk indicators; SD: standard deviation. ** indicates *p* < 0.01.

**Table 7 ijerph-17-03517-t007:** Descriptive statistics and correlation matrix for older workers.

Variables	Mean	SD	HRI	FS	PWB	JS	WS	TP
HRI	–	–	1.00					
FS	3.59	0.87	−0.27 **	1.00				
PWB	3.84	0.82	−0.14	0.20 *	1.00			
JS	3.21	0.90	0.01	0.39 **	0.10	1.00		
WS	3.44	0.82	0.03	0.25 **	0.16 *	0.48 **	1.00	
TP	3.80	0.89	−0.03	0.29 **	0.25 **	0.27 **	0.22 **	1.00

Note: * indicates *p* < 0.05; ** indicates *p* < 0.01.

**Table 8 ijerph-17-03517-t008:** The validity indicator of Group A’s and Group B’s models, and recommended values.

Fit Indexes	Group A ≤45 Years: Young Workers		Group B ≥46 Years: Older Workers		Recommend Value
	Initial	Final	Initial	Final	
χ^2^/df	2.19	1.58	2.36	1.19	1–2 [82]
CFI	0.857	0.960	0.833	0.987	≒1 [83]
TLI	0.832	0.943	0.803	0.982	≒1 [83]
GFI	0.987	0.996	0.990	0.998	≒1 [83]
RMSEA	0.085	0.060	0.093	0.034	Less than 0.08 [79]

CFI: Comparative Fit Index; TLI: Tucker–Lewis index; GFI: Goodness of Fit Index; RMSEA: Root Mean Square Error of Approximation.

**Table 9 ijerph-17-03517-t009:** Direct, indirect, and total effects of path coefficients.

Title	Group A (≤45 Years: Young Workers)			Group B (≥46 Years: Older Workers)		
	Direct	Indirect	Total	Direct	Indirect	Total
HRI → FS	n.s.	–	n.s.	−0.44	–	−0.44
HRI → PWB	n.s.	–	n.s.	−0.23	–	−0.23
FS → TP	0.21	0.04	0.25	0.27	0.06	0.33
PWB → TP	0.45	–	0.45	0.23	–	0.23
HRI → TP	n.s.	–	n.s.	–	−0.19	−0.19

Note: n.s.: not significant.

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
