# Peer review of "The Effects of Psychological Factors on Perceptions of Productivity in Construction Sites in Japan by Worker Age"

_ijerph, 2020, doi:10.3390/ijerph17103517_

Round 1

Reviewer 1 Report

This is a very interesting paper on an important topic – balance between productivity and safety in the construction industry. Construction industry is indeed a major sector in modern economies in term of employment. At the same time, it concentrates a certain number of problems, related to work accidents, eventually leading to death at work. The paper is based on an empirical investigation focusing on one Japanese company. Although this case study may raise some concerns about the degree of generality of the results, I consider this is not at all a problem, given the importance of the Japanese economy and the importance of the construction industry for the Japanese economy. Moreover, the trade-off between safety and productivity has been well documented in the case of the construction industry. Last but not least, the paper proposes a quite original approach – to my limited knowledge – in looking at the relationships between the feeling of safety in the work environment, proactive, work behavior, job satisfaction, work skills, team performance, and health risk indicators such as heart rate, among construction workers of different ages. I am quite impressed by the effort by the authors to build a questionnaire survey for the workers  (n  =  357) of this construction company and to be allowed to conduct and to publish this research.

However, I have some concerns and problems:

  • The major one is related to the theoretical model presented in section 2.3. Obviously, there are too many hypotheses that cannot be considered and tested simultaneously. This is particularly apparent in the figure 2 of the paper, which is very fragile, because of the existence of side and reverse effects, which cannot be dealt with by the authors. To put it shortly, there are some endogeneity problems that are not treated and it should be at least recognized. Therefore, I would be much more modest than them, when they claim, in figures 3 & 4, some results that are in fact much more fragile than the levels of significance of econometric estimations seem to indicate. The authors seem to be partly aware of this major limitation, as it is visible in the conclusion. I would like to advise them to draw some conclusions from these weaknesses and to be much more prudent in the interpretation of their results, to say the least.
  • A minor one is as follows. In order to give more value and visibility to this research, I would advise the authors to have a second proofreading and editing check. It would allow them avoiding some small mistakes (some typos or some problems in the structure of the paper, like the absence of the section 3). It would make the paper more readable (for example, on line 64, I do not understand the meaning of the following sentence “there have been very  few reports of the effects of construction workers on productivity awareness, along with psychological factors and health risk indicators”)

Reviewer 2 Report

The structure of chapters is strange.
Specifically, Chapter 3 does not exist.

2.2
Please discuss and show that it is appropriate to classify workers by age.
It can also be imagined that differences in experience in construction site rather than differences due to age influence safety awareness.

2.3.3
The authors write that "There is no general agreement on defining job satisfaction or occupation yet. Job satisfaction is a complex and multifaceted concept that has different meanings according to different authors.".
On the other hand, as can be seen from Fig. 2, the authors treat "Job Satisfacation" as an evaluation item.
I think there is not enough clear description of the definition of "Job satisfaction" in this paper.

2.5.
There is a complete lack of description of how the workers' heart rate and BMI are parsed.

6.
The authors write that "Our study found that health risk indicators, feeling of safety, job satisfaction, awareness of work skills and proactive work behaviours in the workplace suggested to affect workplace productivity.".
However, there seems to be a lack of discussion about what the results of this research implies workplace productivity in Chapter 5.
